# PICKING WINNING TICKETS BEFORE TRAINING BY PRESERVING GRADIENT FLOW

**Chaoqi Wang, Guodong Zhang, Roger Grosse**
University of Toronto, Vector Institute
`{cqwang, gdzhang, rgrosse}@cs.toronto.edu`

## ABSTRACT

Overparameterization has been shown to benefit both the optimization and generalization of neural networks, but large networks are resource hungry at both training and test time. Network pruning can reduce test-time resource requirements, but is typically applied to trained networks and therefore cannot avoid the expensive training process. We aim to prune networks at initialization, thereby saving resources at training time as well. Specifically, we argue that efficient training requires preserving the gradient flow through the network. This leads to a simple but effective pruning criterion we term Gradient Signal Preservation (GraSP). We empirically investigate the effectiveness of the proposed method with extensive experiments on CIFAR-10, CIFAR-100, Tiny-ImageNet and ImageNet, using VGGNet and ResNet architectures. Our method can prune 80% of the weights of a VGG-16 network on ImageNet at initialization, with only a 1.6% drop in top-1 accuracy. Moreover, our method achieves significantly better performance than the baseline at extreme sparsity levels. Our code is made public at: https://github.com/alecwangcq/GraSP.

## 1 INTRODUCTION

Deep neural networks exhibit good optimization and generalization performance in the overparameterized regime (Zhang et al., 2016; Neyshabur et al., 2019; Arora et al., 2019; Zhang et al., 2019b), but both training and inference for large networks are computationally expensive. Network pruning (LeCun et al., 1990; Hassibi et al., 1993; Han et al., 2015b; Dong et al., 2017; Zeng & Urtasun, 2019; Wang et al., 2019) has been shown to reduce the *test-time* resource requirements with minimal performance degradation. However, as the pruning is typically done to a trained network, these methods don't save resources at *training time*. Moreover, it has been argued that it is hard to train sparse architectures from scratch while maintaining comparable performance to their dense counterparts (Han et al., 2015a; Li et al., 2016). Therefore, we ask: can we prune a network prior to training, so that we can improve computational efficiency at training time?

Recently, Frankle & Carbin (2019) shed light on this problem by proposing the Lottery Ticket Hypothesis (LTH), namely that there exist sparse, trainable sub-networks (called "winning tickets") within the larger network. They identify the winning tickets by taking a *pre-trained* network and removing connections with weights smaller than a pre-specified threshold. They then reset the remaining weights to their initial values, and retrain the sub-network from scratch. Hence, they showed that the pre-trained weights are not necessary, only the pruned architecture and the corresponding initial weight values. Nevertheless, like traditional pruning methods, the LTH approach still requires training the full-sized network in order to identify the sparse sub-networks.

Can we identify sparse, trainable sub-networks at initialization? This would allow us to exploit sparse computation with specified hardware for saving computation cost. (For instance, Dey et al. (2019) demonstrated 5x efficiency gains for training networks with pre-specified sparsity.) At first glance, a randomly initialized network seems to provide little information that we can use to judge the importance of individual connections, since the choice would seem to depend on complicated training dynamics. However, recent work suggests this goal is attainable. Lee et al. (2018) proposed the first algorithm for pruning at initialization time: Single-shot Network Pruning (SNIP), which uses a *connection sensitivity* criterion to prune weights with both small magnitude and small gradients.

Their empirical results are promising in the sense that they can find sparse, trainable sub-networks at initialization. However, connection sensitivity is sub-optimal as a criterion because the gradient of each weight might change dramatically after pruning due to complicated interactions between weights. Since SNIP only considers the gradient for one weight in isolation, it could remove connections that are important to the flow of information through the network. Practically, we find that this blocking of information flow manifests as a reduction in the norm of the gradient.

Therefore, we aim to prune connections in a way that accounts for their role in the network's gradient flow. Specifically, we take the gradient norm *after pruning* as our criterion, and prune those weights whose removal will result in least decrease in the gradient norm after pruning. Because we rely on preserving the gradient flow to prune the network, we name our method Gradient Signal Preservation (GraSP). Our approach is easy to implement and conceptually simple. Moreover, the recently introduced Neural Tangent Kernel (NTK) (Jacot et al., 2018) provides tools for studying the learning dynamics in the output space. Building on the analysis of Arora et al. (2019), we show that our pruning criterion tends to keep those weights which will be beneficial for optimization. We evaluate GraSP on CIFAR-10, CIFAR-100 (Krizhevsky, 2009), Tiny-ImageNet and ImageNet (Deng et al., 2009) with modern neural networks, such as VGGNet (Simonyan & Zisserman, 2014) and ResNet (He et al., 2016). GraSP significantly outperforms SNIP in the extreme sparsity regime.

## 2 RELATED WORK AND BACKGROUND

In this section, we review the literature on neural network pruning including pruning after training, pruning during training, dynamic sparse training and pruning before training. We then discuss propagation of signals in deep neural networks and recent works in dynamical isometry and mean-field theory. Lastly, we also review the Neural Tangent Kernel (NTK) (Jacot et al., 2018), which builds up the foundation for justifying our method in Section 4.

### 2.1 NETWORK PRUNING

**After training.** Most pruning algorithms (LeCun et al., 1990; Hassibi et al., 1993; Dong et al., 2017; Han et al., 2015b; Li et al., 2016; Molchanov et al., 2016) operate on a pre-trained network. The main idea is to identify those weights which are most redundant, and whose removal will therefore least degrade the performance. Magnitude based pruning algorithms (Han et al., 2015b;a) remove those weights which are smaller than a threshold, which may incorrectly measure the importance of each weight. In contrast, Hessian-based pruning algorithms (LeCun et al., 1990; Hassibi et al., 1993) compute the importance of each weight by measuring how its removal will affect the loss. More recently, Wang et al. (2019) proposed a network reparameterization based on the Kronecker-factored Eigenbasis for further boosting the performance of Hessian-based methods. However, all the aforementioned methods require pre-training, and therefore aren't applicable at initialization.

**During training.** There are also some works which attempt to incorporate pruning into the training procedure itself. Srinivas & Babu (2016) proposed generalized dropout, allowing for tuning the individual dropout rates during training, which can result in a sparse network after training. Louizos et al. (2018) proposed a method for dealing with discontinuity in training $L_0$ norm regularized networks in order to obtain sparse networks. Both methods require roughly the same computational cost as training the full network.

**Dynamic Sparse Training** Another branch of pruning algorithms is Dynamic Sparse Training methods, which will dynamically change the weight sparsity during training. Representative works, such as Bellec et al. (2018); Mocanu et al. (2018); Mostafa & Wang (2019); Dettmers & Zettlemoyer (2019), follow a prune-redistribute-regrowth cycle for pruning. Among them, Dettmers & Zettlemoyer (2019), proposed the sparse momentum algorithm, which dynamically determines the sparse mask based on the mean momentum magnitude during training. However, their method requires maintaining the momentum of *all* the weights during training, and thus does not save memory. These techniques generally achieve higher accuracy compared with fixed sparse connectivity, but change the standard training procedure and therefore do not enjoy the potential hardware acceleration.

**Before training.** Pruning at initialization is more challenging because we need to account for the effect on the training dynamics when removing each weight. There have been several attempts to conduct pruning before training. Frankle & Carbin (2019); Frankle et al. (2019) proposed and

validated the Lottery Ticket Hypothesis (LTH), namely that the network structure found by traditional pruning algorithms and the corresponding initialization are together sufficient for training the sub-network from scratch. Lee et al. (2018) proposed the SNIP algorithm, which was the first attempt to directly identify trainable and sparse sub-networks at initialization time. Their method was based on *connection sensitivity*, which aims to preserve the loss after pruning, and achieved impressive results. Concurrently to our work, Lee et al. (2019b) studied the pruning problem from a signal propagation perspective, and proposed to use an orthogonal initialization to ensure faithful signal propagation. Though their work shares the same spirit as our GraSP algorithm, they focused on the weight initialization scheme, while we focus on the pruning criterion.

## 2.2 Signal Propagation at Initialization

Our pruning criteria shares the same spirit as recent works in dynamical isometry and mean-field theory (Saxe et al., 2013; Xiao et al., 2018; Yang & Schoenholz, 2017; Poole et al., 2016) where they derived initialization scheme theoretically by developing a mean field theory for signal propagation and by characterizing singular values of the input-output Jacobian matrix. Particularly, Xiao et al. (2018) successfully train 10,000-layers vanilla ConvNets with specific initialization scheme. Essentially, this line of work shows that the trainability of a neural network at initialization is crucial for final performance and convergence. While they focus on address the trainability issue of very deep networks, we aim to solve the issue of sparse neural networks. Besides, they measure the trainability by examining the input-output Jacobian while we do that by checking the gradient norm. Though different, gradient norm is closely related to the input-output Jacobian.

## 2.3 Neural Tangent Kernel and Convergence Analysis

Jacot et al. (2018) analyzed the dynamics of neural net training by directly analyzing the evolution of the network's predictions in output space. Let $\mathcal{L}$ denote the cost function, $\mathcal{X}$ the set of all training samples, $\mathcal{Z} = f(\mathcal{X}; \boldsymbol{\theta}) \in \mathbb{R}^{nk \times 1}$ the outputs of the neural network, and $k$ and $n$ the output space dimension and the number of training examples. For a step of gradient descent, the change to the network's predictions can be approximated with a first-order Taylor approximation:

$$f(\mathcal{X}; \boldsymbol{\theta}_{t+1}) = f(\mathcal{X}; \boldsymbol{\theta}_t) - \eta \boldsymbol{\Theta}_t(\mathcal{X}, \mathcal{X}) \nabla_{\mathcal{Z}} \mathcal{L}, \tag{1}$$

where the matrix $\boldsymbol{\Theta}_t(\mathcal{X}, \mathcal{X})$ is the *Neural Tangent Kernel (NTK)* at time step $t$:

$$\boldsymbol{\Theta}_t(\mathcal{X}, \mathcal{X}) = \nabla_{\boldsymbol{\theta}} f(\mathcal{X}; \boldsymbol{\theta}_t) \nabla_{\boldsymbol{\theta}} f(\mathcal{X}; \boldsymbol{\theta}_t)^\top \in \mathbb{R}^{nk \times nk}, \tag{2}$$

where $\nabla_{\boldsymbol{\theta}} f(\mathcal{X}; \boldsymbol{\theta})$ denotes the network Jacobian over the whole training set. Jacot et al. (2018) showed that for infinitely wide networks, with proper initialization, the NTK exactly captures the output space dynamics throughout training. In particular, $\boldsymbol{\Theta}_t(\mathcal{X}, \mathcal{X})$ remains constant throughout training. Arora et al. (2019) used the NTK to analyze optimization and generalization phenomena, showing that under the assumptions of constant NTK and squared error loss, the training dynamics can be analyzed in closed form:

$$\|\mathcal{Y} - f(\mathcal{X}; \boldsymbol{\theta}_t)\|_2 = \sqrt{\sum_{i=1}^{n} (1 - \eta \lambda_i)^{2t} (\mathbf{u}_i^\top \mathcal{Y})^2} \pm \epsilon \tag{3}$$

where $\mathcal{Y} \in \mathbb{R}^{nk \times 1}$ is all the targets, $\boldsymbol{\Theta} = \mathbf{U} \boldsymbol{\Lambda} \mathbf{U}^\top = \sum_{i=1}^{n} \lambda_i \mathbf{u}_i \mathbf{u}_i^\top$ is the eigendecomposition, and $\epsilon$ is a bounded error term. Although the constant NTK assumption holds only in the infinite width limit, Lee et al. (2019a) found close empirical agreement between the NTK dynamics and the true dynamics for wide but practical networks, such as wide ResNet architectures (Zagoruyko & Komodakis, 2016).

## 3 Revisiting Single-Shot Network Pruning (SNIP)

Single-shot network pruning was introduced by Lee et al. (2018), who used the term to refer both to the general problem setting and to their specific algorithm. To avoid ambiguity, we refer to the general problem of pruning before training as *foresight pruning*. For completeness, we first revisit the formulation of foresight pruning, and then point out issues of SNIP for motivating our method.

---

**Algorithm 1** Gradient Signal Preservation (GraSP).

---

**Require:** Pruning ratio $p$, training data $\mathcal{D}$, network $f$ with initial parameters $\boldsymbol{\theta}_0$
  1: $\mathcal{D}_b = \{(\mathbf{x}_i, \mathbf{y}_i)\}_{i=1}^b \sim \mathcal{D}$                  ▷ Sample a collection of training examples
  2: Compute the Hessian-gradient product $\mathbf{Hg}$ (see Eqn. (8))           ▷ See Algorithm 2
  3: $\mathbf{S}(-\boldsymbol{\theta}_0) = -\boldsymbol{\theta}_0 \odot \mathbf{Hg}$                ▷ Compute the importance of each weight
  4: Compute $p_{\text{th}}$ percentile of $\mathbf{S}(-\boldsymbol{\theta}_0)$ as $\tau$
  5: $\mathbf{m} = \mathbf{S}(-\boldsymbol{\theta}_0) < \tau$              ▷ Remove the weights with smallest importance
  6: Train the network $f_{\mathbf{m}\odot\boldsymbol{\theta}}$ on $\mathcal{D}$ until convergence.

---

**Problem Formulation.** Suppose we have a neural network $f$ parameterized by $\boldsymbol{\theta} \in \mathbb{R}^d$, and our objective is to minimize the empirical risk $\mathcal{L}(\boldsymbol{\theta}) = \frac{1}{N}\sum_i [\ell(f(\mathbf{x}_i; \boldsymbol{\theta}), y_i)]$ given a training set $\mathcal{D} = \{(\mathbf{x}_i, y_i)\}_{i=1}^N$. Then, the foresight pruning problem can be formulated as:

$$\min_{\mathbf{m}\in\{0,1\}^d} \mathbb{E}_{(\mathbf{x},y)\sim\mathcal{D}} \left[\ell\left(f\left(\mathbf{x}; \mathcal{A}(\mathbf{m}, \boldsymbol{\theta}_0)\right), y\right)\right] \quad \text{s.t.} \quad \|\mathbf{m}\|_0/d = 1 - p \tag{4}$$

where $\lceil p \cdot d \rceil$ is the number of weights to be removed, and $\mathcal{A}$ is a known training algorithm (e.g. SGD), which takes the mask $\mathbf{m}$ (here we marginalize out the initial weights $\boldsymbol{\theta}_0$ for simplicity), and returns the trained weights. Since globally minimizing Eqn. 4 is intractable, we are instead interested in heuristics that result in good practical performance.

**Revisiting SNIP.** SNIP (Lee et al., 2018) was the first algorithm proposed for foresight pruning, and it leverages the notion of *connection sensitivity* to remove unimportant connections. They define this in terms of how removing a single weight $\theta_q$ in isolation will affect the loss:

$$S(\theta_q) = \lim_{\epsilon\to 0} \left|\frac{\mathcal{L}(\boldsymbol{\theta}_0) - \mathcal{L}(\boldsymbol{\theta}_0 + \epsilon\boldsymbol{\delta}_q)}{\epsilon}\right| = \left|\theta_q \frac{\partial\mathcal{L}}{\partial\theta_q}\right| \tag{5}$$

where $\theta_q$ is the $q_{th}$ element of $\boldsymbol{\theta}_0$, and $\boldsymbol{\delta}_q$ is a one-hot vector whose $q_{th}$ element equals $\theta_q$. Essentially, SNIP aims to preserve the loss of the original randomly initialized network.

Preserving the loss value motivated several classic methods for pruning a *trained network*, such as optimal brain damage (LeCun et al., 1990) and optimal brain surgery (Hassibi et al., 1993). While the motivation for loss preservation of a trained network is clear, it is less clear why this is a good criterion for foresight pruning. After all, at initialization, the loss is no better than chance. We argue that at the *beginning* of training, it is more important to preserve the training dynamics than the loss itself. SNIP does not do this automatically, because even if removing a particular connection doesn't affect the loss, it could still block the flow of information through the network. For instance, we noticed in our experiments that SNIP with a high pruning ratio (e.g. 99%) tends to eliminate nearly all the weights in a particular layer, creating a bottleneck in the network. Therefore, we would prefer a pruning criterion which accounts for how the presence or absence of one connection influences the training of the rest of the network.

## 4   Gradient Signal Preservation

We now introduce and motivate our foresight pruning criterion, Gradient Signal Preservation (GraSP). To understand the problem we are trying to address, observe that the network after pruning will have fewer parameters and sparse connectivity, hindering the flow of gradients through the network and potentially slowing the optimization. This is reflected in Figure 2, which shows the reduction in gradient norm for random pruning with various pruning ratios. Moreover, the performance of the pruned networks is correspondingly worse (see Table 1).

Mathematically, a larger gradient norm indicates that, to the first order, each gradient update achieves a greater loss reduction, as characterized by the following directional derivative:

$$\Delta\mathcal{L}(\boldsymbol{\theta}) = \lim_{\epsilon\to 0} \frac{\mathcal{L}(\boldsymbol{\theta} + \epsilon\nabla\mathcal{L}(\boldsymbol{\theta})) - \mathcal{L}(\boldsymbol{\theta})}{\epsilon} = \nabla\mathcal{L}(\boldsymbol{\theta})^\top \nabla\mathcal{L}(\boldsymbol{\theta}) \tag{6}$$

We would like to preserve or even increase (if possible) the gradient flow *after pruning* (*i.e.*, the gradient flow of the pruned network). Following LeCun et al. (1990), we cast the pruning operation

---

**Algorithm 2** Hessian-gradient Product.

---

**Require:** A batch of training data $\mathcal{D}_b$, network $f$ with initial parameters $\boldsymbol{\theta}_0$, loss function $\mathcal{L}$
1: $\mathcal{L}(\boldsymbol{\theta}_0) = \mathbb{E}_{(\mathbf{x},y)\sim\mathcal{D}_b}[\ell(f(\mathbf{x};\boldsymbol{\theta}_0),y)]$      ▷ Compute the loss and build the computation graph
2: $\mathbf{g} = \mathrm{grad}(\mathcal{L}(\boldsymbol{\theta}_0),\boldsymbol{\theta}_0)$      ▷ Compute the gradient of loss function with respect to $\boldsymbol{\theta}_0$
3: $\mathbf{Hg} = \mathrm{grad}(\mathbf{g}^\top \mathrm{stop\_grad}(\mathbf{g}),\boldsymbol{\theta}_0)$      ▷ Compute the Hessian vector product of $\mathbf{Hg}$
4: Return $\mathbf{Hg}$

---

as adding a perturbation $\boldsymbol{\delta}$ to the initial weights. We then use a Taylor approximation to characterize how removing one weight will affect the gradient flow *after pruning*:

$$\mathbf{S}(\boldsymbol{\delta}) = \Delta\mathcal{L}(\boldsymbol{\theta}_0+\boldsymbol{\delta}) - \underbrace{\Delta\mathcal{L}(\boldsymbol{\theta}_0)}_{\text{Const}} = 2\boldsymbol{\delta}^\top\nabla^2\mathcal{L}(\boldsymbol{\theta}_0)\nabla\mathcal{L}(\boldsymbol{\theta}_0) + \mathcal{O}(\|\boldsymbol{\delta}\|_2^2)$$

$$= 2\boldsymbol{\delta}^\top\mathbf{Hg} + \mathcal{O}(\|\boldsymbol{\delta}\|_2^2), \quad (7)$$

where $\mathbf{S}(\boldsymbol{\delta})$ approximately measures the change to (6). The Hessian matrix $\mathbf{H}$ captures the dependencies between different weights, and thus helps predict the effect of removing multiple weights. When $\mathbf{H}$ is approximated as the identity matrix, the above criterion recovers SNIP up to the absolute value (recall the SNIP criterion is $|\boldsymbol{\delta}^\top\mathbf{g}|$). However, it has been observed that different weights are highly coupled (Hassibi et al., 1993), indicating that $\mathbf{H}$ is in fact far from the identity.

GraSP uses eqn. (7) as the measure of the importance of each weight. Specifically, if $S(\boldsymbol{\delta})$ is negative, then removing the corresponding weights will reduce the gradient flow, while if it is positive, it will increase the gradient flow. We prefer to first remove those weights whose removal will not reduce the gradient flow. For each weight, the importance can be computed in the following way (by an abuse of notation, we use bold $\mathbf{S}$ to denote vectorized importance):

$$\mathbf{S}(-\boldsymbol{\theta}) = -\boldsymbol{\theta}\odot\mathbf{Hg} \quad (8)$$

For a given pruning ratio $p$, we obtain the resulting pruning mask by computing the importance score of every weight, and removing the bottom $p$ fraction of the weights (see Algorithm 1). Hence, GraSP takes the gradient flow into account for pruning. GraSP is efficient and easy to implement; the Hessian-gradient product can be computed without explicitly constructing the Hessian using higher-order automatic differentiation (Pearlmutter, 1994; Schraudolph, 2002).

### 4.1 UNDERSTANDING GRASP THROUGH LINEARIZED TRAINING DYNAMICS

The above discussion concerns only the training dynamics at initialization time. To understand the longer-term dynamics, we leverage the recently proposed Neural Tangent Kernel (NTK), which has been shown to be able to capture the training dynamics throughout training for practical networks (Lee et al., 2019a). Specifically, as we introduced in section 2.3, eqn. (3) characterizes how the training error changes throughout the training process, which only depends on time step $t$, training targets $\mathcal{Y}$ and the NTK $\boldsymbol{\Theta}$. Since the NTK stays almost constant for wide but practical networks (Lee et al., 2019a), *e.g.*, wide ResNet (Zagoruyko & Komodakis, 2016), eqn. 3 is fairly accurate in those settings. This shows that NTK captures the training dynamics throughout the training process. Now we decompose eqn. (6) in the following form:

$$\nabla\mathcal{L}(\boldsymbol{\theta})^\top\nabla\mathcal{L}(\boldsymbol{\theta}) = \nabla_\mathcal{Z}\mathcal{L}^\top\boldsymbol{\Theta}(\mathcal{X},\mathcal{X})\nabla_\mathcal{Z}\mathcal{L} = (\mathbf{U}^\top\nabla_\mathcal{Z}\mathcal{L})^\top\boldsymbol{\Lambda}(\mathbf{U}^\top\nabla_\mathcal{Z}\mathcal{L}) = \sum_{i=1}^n\lambda_i(\mathbf{u}_i^\top\mathcal{Y})^2 \quad (9)$$

By relating it to eqn. (3), we can see that GraSP implicitly encourages $\boldsymbol{\Theta}$ to be large in the directions corresponding to output space gradients. Since larger eigenvalue directions of $\boldsymbol{\Theta}$ train faster according to eqn. (3), this suggests that GraSP should result in efficient training. In practice, the increasing of gradient norm might be achieved by increasing the loss. Therefore, we incorporate a temperature term on the logits to smooth the prediction, and so as to reduce the effect introduced by the loss.

## 5 EXPERIMENTS

In this section, we conduct various experiments to validate the effectiveness of our proposed single-shot pruning algorithm in terms of the test accuracy vs. pruning ratios by comparing against

**Table 1:** Comparisons with Random Pruning with VGG19 and ResNet32 on CIFAR-10/100.

| Dataset | CIFAR-10 | | | | CIFAR-100 | | | |
|---|---|---|---|---|---|---|---|---|
| Pruning ratio | 95% | 98% | 99% | 99.5% | 95% | 98% | 99% | 99.5% |
| Random Pruning (VGG19) | 89.47(0.5) | 86.71(0.7) | 82.21(0.5) | 72.89(1.6) | 66.36(0.3) | 61.33(0.1) | 55.18(0.6) | 36.88(6.8) |
| GraSP (VGG19) | **93.04(0.2)** | **92.19(0.1)** | **91.33(0.1)** | **88.61(0.7)** | **71.23(0.1)** | **68.90(0.5)** | **66.15(0.2)** | **60.21(0.1)** |
| Random Pruning (ResNet32) | 89.75(0.1) | 85.90(0.4) | 71.78(9.9) | 50.08(7.0) | 64.72(0.2) | 50.92(0.9) | 34.62(2.8) | 18.51(0.43) |
| GraSP (ResNet32) | **91.39(0.3)** | **88.81(0.1)** | **85.43(0.5)** | **80.50(0.3)** | **66.50(0.1)** | **58.43(0.4)** | **48.73(0.3)** | **35.55(2.4)** |

**Table 2:** Test accuracy of pruned VGG19 and ResNet32 on CIFAR-10 and CIFAR-100 datasets. The bold number is the higher one between the accuracy of GraSP and that of SNIP.

| Dataset | CIFAR-10 | | | CIFAR-100 | | |
|---|---|---|---|---|---|---|
| Pruning ratio | 90% | 95% | 98% | 90% | 95% | 98% |
| **VGG19** (Baseline) | 94.23 | - | - | 74.16 | - | - |
| OBD (LeCun et al., 1990) | 93.74 | 93.58 | 93.49 | 73.83 | 71.98 | 67.79 |
| MLPrune (Zeng & Urtasun, 2019) | 93.83 | 93.69 | 93.49 | 73.79 | 73.07 | 71.69 |
| LT (original initialization) | 93.51 | 92.92 | 92.34 | 72.78 | 71.44 | 68.95 |
| LT (reset to epoch 5) | 93.82 | 93.61 | 93.09 | 74.06 | 72.87 | 70.55 |
| DSR (Mostafa & Wang, 2019) | 93.75 | 93.86 | 93.13 | 72.31 | 71.98 | 70.70 |
| SET Mocanu et al. (2018) | 92.46 | 91.73 | 89.18 | 72.36 | 69.81 | 65.94 |
| Deep-R (Bellec et al., 2018) | 90.81 | 89.59 | 86.77 | 66.83 | 63.46 | 59.58 |
| SNIP (Lee et al., 2018) | **93.63±0.06** | **93.43±0.20** | 92.05±0.28 | **72.84±0.22** | **71.83±0.23** | 58.46±1.10 |
| GraSP | 93.30±0.14 | 93.04±0.18 | **92.19±0.12** | 71.95±0.18 | 71.23±0.12 | **68.90±0.47** |
| **ResNet32** (Baseline) | 94.80 | - | - | 74.64 | - | - |
| OBD (LeCun et al., 1990) | 94.17 | 93.29 | 90.31 | 71.96 | 68.73 | 60.65 |
| MLPrune (Zeng & Urtasun, 2019) | 94.21 | 93.02 | 89.65 | 72.34 | 67.58 | 59.02 |
| LT (original initialization) | 92.31 | 91.06 | 88.78 | 68.99 | 65.02 | 57.37 |
| LT (reset to epoch 5) | 93.97 | 92.46 | 89.18 | 71.43 | 67.28 | 58.95 |
| DSR (Mostafa & Wang, 2019) | 92.97 | 91.61 | 88.46 | 69.63 | 68.20 | 61.24 |
| SET Mocanu et al. (2018) | 92.30 | 90.76 | 88.29 | 69.66 | 67.41 | 62.25 |
| Deep-R (Bellec et al., 2018) | 91.62 | 89.84 | 86.45 | 66.78 | 63.90 | 58.47 |
| SNIP (Lee et al., 2018) | **92.59±0.10** | 91.01±0.21 | 87.51±0.31 | 68.89±0.45 | 65.22±0.69 | 54.81±1.43 |
| GraSP | 92.38±0.21 | **91.39±0.25** | **88.81±0.14** | **69.24±0.24** | **66.50±0.11** | **58.43±0.43** |

SNIP. We also include three Dynamic Sparse Training methods, DSR (Mostafa & Wang, 2019), SET (Mocanu et al., 2018) and Deep-R (Bellec et al., 2018). We further include two traditional pruning algorithms (LeCun et al., 1990; Zeng & Urtasun, 2019), which operate on the pre-trained networks, for serving as an upper bound for foresight pruning. Besides, we study the convergence performance of the sub-networks obtained by different pruning methods for investigating the relationship between gradient norm and final performance. Lastly, we study the role of initialization and batch sizes in terms of the performance of GraSP for ablation study.

## 5.1 Pruning Results on Modern ConvNets

To evaluate the effectiveness of GraSP on real world tasks, we test GraSP on four image classification datasets, CIFAR-10/100, Tiny-ImageNet and ImageNet, with two modern network architectures, VGGNet and ResNet[1]. For the experiments on CIFAR-10/100 and Tiny-ImageNet, we use a mini-batch with ten times of the number of classes for both GraSP and SNIP[2] according to Lee et al. (2018). The pruned network is trained with Kaiming initialization (He et al., 2015) using SGD for 160 epochs for CIFAR-10/100, and 300 epochs for Tiny-ImageNet, with an initial learning rate of 0.1 and batch size 128. The learning rate is decayed by a factor of 0.1 at 1/2 and 3/4 of the total number of epochs. Moreover, we run each experiment for 3 trials for obtaining more stable results. For ImageNet, we adopt the Pytorch (Paszke et al., 2017) official implementation, but we used more epochs for training according to Liu et al. (2019). Specifically, we train the pruned networks with SGD for 150 epochs, and decay the learning rate by a factor of 0.1 every 50 epochs.

We first compre GraSP against random pruning, which generates the mask randomly for a given pruning ratio. The test accuracy is reported in Table 1. We can observe GraSP clearly outperforms

---

[1]For experiments on CIFAR-10/100 and Tiny-ImageNet, we adopt ResNet32 and double the number of filters in each convolutional layer for making it able to overfit CIFAR-100.

[2]For SNIP, we adopt the implementation public at https://github.com/mi-lad/snip

**Table 4:** Test accuracy of pruned VGG19 and ResNet32 on Tiny-ImageNet dataset. The bold number is the higher one between the accuracy of GraSP and that of SNIP.

| Network | VGG19 | | | ResNet32 | | |
|---|---|---|---|---|---|---|
| Pruning ratio | 90% | 95% | 98% | 85% | 90% | 95% |
| **VGG19/ResNet32** (Baseline) | 61.38 | - | - | 62.89 | - | - |
| OBD (LeCun et al., 1990) | 61.21 | 60.49 | 54.98 | 58.55 | 56.80 | 51.00 |
| MLPrune (Zeng & Urtasun, 2019) | 60.23 | 59.23 | 55.55 | 58.86 | 57.62 | 51.70 |
| LT (original initialization) | 60.32 | 59.48 | 55.12 | 56.52 | 54.27 | 49.47 |
| LT (reset to epoch 5) | 61.19 | 60.57 | 56.18 | 60.31 | 57.77 | 51.21 |
| DSR (Mostafa & Wang, 2019) | 62.43 | 59.81 | 58.36 | 57.08 | 57.19 | 56.08 |
| SET Mocanu et al. (2018) | 62.49 | 59.42 | 56.22 | 57.02 | 56.92 | 56.18 |
| Deep-R (Bellec et al., 2018) | 55.64 | 52.93 | 49.32 | 53.29 | 52.62 | 52.00 |
| SNIP (Lee et al., 2018) | **61.02±0.41** | 59.27±0.39 | 48.95±1.73 | 56.33±0.24 | 55.43±0.14 | 49.57±0.44 |
| GraSP | 60.76±0.23 | **59.50±0.33** | **57.28±0.34** | **57.25±0.11** | **55.53±0.11** | **51.34±0.29** |

random pruning, and the performance gap can be up to more than $30\%$ in terms of test accuracy. We further compare GraSP with more competitive baselines on CIFAR-10/100 and Tiny-ImageNet for pruning ratios $\{85\%, 90\%, 95\%, 98\%\}$, and the results can be referred in Table 2 and 4. When the pruning ratio is low, i.e. $85\%, 90\%$, both SNIP and GraSP can achieve very close results to the baselines, though still do not outperform pruning algorithms that operate on trained networks, as expected. However, GraSP achieves significant better results for higher pruning ratios with more complicated networks (e.g. ResNet) and datasets (e.g. CIFAR-100 and Tiny-ImageNet), showing the advantages of directly relating the pruning criteria with the gradient flow after pruning. Moreover, in most cases, we notice that either SNIP or GraSP can match or even slightly outperform LT with original initialization, which indicates that both methods can identify meaningful structures. We further experiment with the late resetting as suggested in Frankle et al. (2019) by resetting the weights of the winning tickets to their values at 5 epoch. By doing so, we observe a boost in the performance of the LT across all the settings, which is consistent with (Frankle et al., 2019). As for the comparisons with Dynamic Sparse Training (DST) methods, DSR is the best performing one, and also we can observe that GraSP is still quite competitive to them even though without the flexibility to change the sparsity during training. In particular, GraSP can outperform Deep-R in almost all the settings, and surpasses SET in more than half of the settings.

**Table 3:** Test accuracy of ResNet-50 and VGG16 on ImageNet with pruning ratios 60%, 80% and 90%.

| Pruning ratios | 60% | | 80% | | 90% | |
|---|---|---|---|---|---|---|
| Accuracy | top-1 | top-5 | top-1 | top5 | top-1 | top5 |
| **ResNet-50** (Baseline) | 75.70 | 92.81 | - | - | - | - |
| SNIP (Lee et al., 2018) | 73.95 | **91.97** | 69.67 | 89.24 | 61.97 | 82.90 |
| GraSP | **74.02** | 91.86 | **72.06** | **90.82** | **68.14** | **88.67** |
| **VGG16** (Baseline) | 73.37 | 91.47 | - | - | - | - |
| SNIP (Lee et al., 2018) | **72.95** | **91.39** | 69.96 | 89.71 | 65.27 | 86.14 |
| GraSP | 72.91 | 91.18 | **71.65** | **90.58** | **69.94** | **89.48** |

However, the above three datasets are overly simple and small-scale to draw robust conclusions, and thus we conduct further experiments on ImageNet using ResNet-50 and VGG16 with pruning ratios $\{60\%, 80\%, 90\%\}$ to validate the effectiveness of GraSP. The results are shown in Table 3, we can see that when the pruning ratio is $60\%$, both SNIP and GraSP can achieve very close performance to the original one, and these two methods perform almost the same. However, as we increase the pruning ratio, we observe that GraSP surpasses SNIP more and more. When the pruning ratio is $90\%$, GraSP can beat SNIP by $6.2\%$ for ResNet-50 and $4.7\%$ for VGG16 in top-1 accuracy, showing the advantages of GraSP at larger pruning ratios. We seek to investigate the reasons behind in Section 5.2 for obtaining a better understanding on GraSP.

## 5.2 ANALYSIS ON CONVERGENCE PERFORMANCE AND GRADIENT NORM

Based on our analysis in Section 4.1 and the observations in the previous subsection, we conduct further experiments on CIFAR-100 with ResNet32 to investigate where the performance gains come from in the high sparsity regions. We present the training statistics in terms of the training and test loss in Figure 1. We observe that the main bottleneck of pruned neural networks is underfitting, and thus support our optimization considerations when deriving the pruning criteria. As a result, the network pruned with GraSP can achieve much lower loss for both training and testing and the decrease in training loss is also much faster than SNIP.

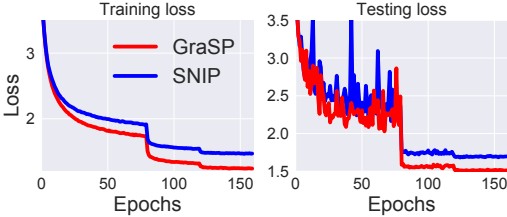
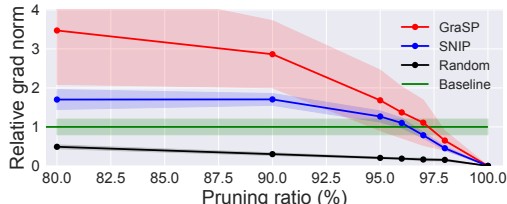

**Figure 1:** The training and testing loss on CIFAR-100 of SNIP and GraSP with ResNet32 and a pruning ratio of 98%.

**Figure 2:** The gradient norm of ResNet32 after pruning on CIFAR-100 of various pruning ratios. Shaded area is the 95% confidence interval calculated with 10 trials.

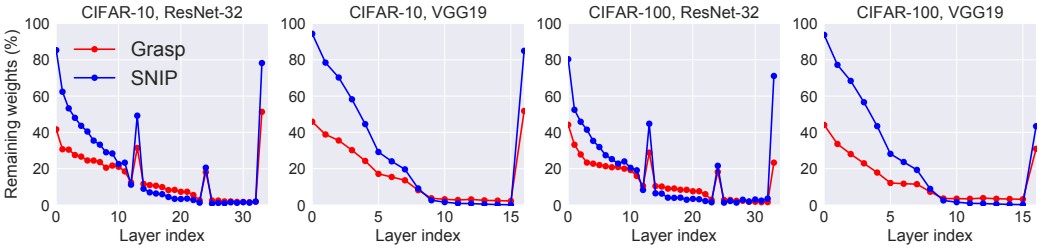

**Figure 3:** The portion of remaining weights at each layer after pruning with a pruning ratio of 95%.

Besides, we also plot the gradient norm of pruned networks at initialization for a ResNet32 on CIFAR-100. The gradient norm is computed as the average of the gradients over the entire dataset, and we normalize the gradient of the original network to be 1. We run each experiment for 10 trials in order to obtain more stable results. We observe that both SNIP and GraSP result in a lower gradient norm at high sparsity (e.g. 98%), but GraSP can better preserve the gradient norm after pruning, and also yield better results than SNIP at the high sparsity regions. Moreover, in these high sparsity regions, the pruned network usually underfits the training data, and thus the optimization will be a problem. The randomly pruned network has a much lower gradient norm and performs the worst. In contrast, the network pruned by GraSP can start with a higher gradient norm and thus hopefully more training progress can be made as evidenced by our results in Figure 1.

## 5.3 VISUALIZING THE NETWORK AFTER PRUNING

In order to probe the difference of the network pruned by SNIP and GraSP, we present the portion of the remaining weights at each layer of the sparse network obtained by SNIP and GraSP in Figure. 3. We observe that these two pruning methods result in different pruning strategies. Specifically, GraSP aims for preserving the gradient flow after pruning, and thus will not prune too aggressively for each layer. Moreover, it has been known that those convolutional layers at the top usually learn highly sparse features and thus more weights can be pruned (Zhang et al., 2019a). As a result, both methods prune most of the weights at the top layers, but GraSP will preserve more weights than SNIP due to the consideration of preserving the gradient flow. In contrast, SNIP is more likely to prune nearly all the weights in top layers, and thus those layers will be the bottleneck of blocking the information flow from the output layer to the input layer.

## 5.4 EFFECT OF BATCH SIZE AND INITIALIZATION

We also study how the batch size and initialization will affect GraSP on CIFAR-10 and CIFAR-100 with ResNet32 for showing the robustness to different hyperparameters. Specifically, we test GraSP with three different initialization methods: Kaiming normal (He et al., 2015), Normal $\mathcal{N}(0, 0.1)$, and Xavier normal (Glorot & Bengio, 2010), as well as different mini-batch sizes. We present the mean and standard variance

**Table 5:** Mean and standard variance of the test accuracy on CIFAR-10 and CIFAR-100 with ResNet32.

| Dataset | CIFAR-10 | | CIFAR-100 | |
|---|---|---|---|---|
| | 60% | 90% | 60% | 90% |
| Kaiming | $93.42 \pm 0.39$ | $92.12 \pm 0.39$ | $71.60 \pm 0.65$ | $68.93 \pm 0.36$ |
| Normal | $93.31 \pm 0.36$ | $92.13 \pm 0.36$ | $71.48 \pm 0.60$ | $67.98 \pm 0.83$ |
| Xavier | $93.32 \pm 0.25$ | $92.22 \pm 0.50$ | $71.10 \pm 1.27$ | $68.11 \pm 0.93$ |

of the test accuracy obtained with different initialization methods and by varying the batch sizes in Table 5. For CIFAR-10 and CIFAR-100, we use batch sizes $\{100, 400, \cdots, 25600, 50000\}$ and $\{1000, 4000, 16000, 32000, 50000\}$ respectively. We observe that GraSP can achieve reasonable performance with different initialization methods, and also the effect of batch size is minimal for networks pruned with Kaiming initialization, which is one of the most commonly adopted initialization techniques in training neural networks.

## 6 DISCUSSION AND CONCLUSION

We propose Gradient Signal Preservation (GraSP), a pruning criterion motivated by preserving the gradient flow through the network after pruning. It can also be interpreted as aligning the large eigenvalues of the Neural Tangent Kernel with the targets. Empirically, GraSP is able to prune the weights of a network *at initialization*, while still performing competitively to traditional pruning algorithms, which requires first training the network. More broadly, foresight pruning can be a way for enabling training super large models that no existing GPUs can fit in, and also reduce the training cost by adopting sparse matrices operations. Readers may notice that there is still a performance gap between GraSP and traditional pruning algorithms, and also the LT with late resetting performs better than LT with original initialization. However, this does not negate the possibility that foresight pruning can match the performance of traditional pruning algorithms while still enjoying cheaper training cost. As evidence, Evci et al. (2019) show that there exists a linear and monotonically decreasing path from the sparse initialization to the solution found by pruning the fully-trained dense network, but current optimizers fail to achieve this. Therefore, designing better gradient-based optimizers that can exploit such paths will also be a good direction to explore.

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
