# OpenReview forum: "Picking Winning Tickets Before Training by Preserving Gradient Flow"
_ICLR.cc/2020/Conference — Accept (Poster)_

### Official Review · AnonReviewer2 · 2019-10-20
**Official Blind Review #2**

**Rating:** 6

**Review:**


The paper proposes a new prunning criterion that performs better than Single-shot Network Pruning (SNIP) in prunning a network at the initalization. This is an important and potentially very impactful research direction, The key idea is to optimize the mask for the loss decrease after an infinimitesal step, rather than for the preservation of loss after prunning. While with the benefit of hindsights it might seem simple, it is a clever innovation. However, I am not convinced by the theoretical explanation and some of the experimental results (see detailed comment below). Based on this I am leaning at the moment towards rejecting the paper. I will be happy to revisit my score if these concerns are addressed.

Detailed comments:

1. I am not sure that NTK based analysis helps explain the efficacy of the method. An increase of the (matrix) norm of the NTK kernel can be achieved by simply scaling up by a constant scalar the logits weights (see for instance https://arxiv.org/abs/1901.08244). Or equivalently (comparing the resulting learning dynamics in NTK, as also can be read from (3)), by just increasing the learning rate. In other words, I could just prune weights randomly, and then scale up logits' weights, and end up with the same effect on the NTK kernel. I think that for this argument to work, NTK kernel should change in a scale-invariant manner. This would correspond to a better conditioning of the loss surface (because Hessian has the same eigenspectrum as the NTK kernel under the NTK assumption), which is a scale invariant property.

2. From the Figure 2 it seems SNIP-prunned network underfits data severly. Could you add training accuracy to the Tables (maybe in the Supplement)? If in all cases when GraSP wins, it is due to underfitting, this should be commented on. Is it common for prunning algorithms to result in underfitting, or is achieving generalization a larger challenge? Could the bad performance at high prunning ratios of SNIP be due to a conflation of two effects: (1) "good" prunning, but (2) lowering the effective learning rate (given the gradient norm is low)? Would, for high prunning ratios, a tuned learning rate improve SNIP performance/reduce underfitting?

3. In Table 5 is the batch-size used for training of the network, or only for the computation of the Hessian-vector product in the GraSP procedure? If for training, then the relatively small spread of results is a bit surprising given results by Keskar (https://arxiv.org/abs/1609.04836)

Edit

Thank you for the rebuttal. Raise my score. I agree with Reviewer #4 that increasing gradient norm at initialization is a promising direction on its own, which warrants acceptance.



**Experience Assessment:**

I have published one or two papers in this area.

**Review Assessment: Checking Correctness Of Derivations And Theory:**

I carefully checked the derivations and theory.

**Review Assessment: Checking Correctness Of Experiments:**

I assessed the sensibility of the experiments.

**Review Assessment: Thoroughness In Paper Reading:**

I read the paper at least twice and used my best judgement in assessing the paper.

---

> ### Author Response · Authors · 2019-11-08
> **Response to reviewer #2**
>
> Thank you for your detailed comments! It’s really encouraging that you think the research direction we’re working on is important.
>
> In terms of your concerns, we address them one by one below.
>
> (1) We agree that scaling up the logits weights leads to the same effect on the NTK. Nevertheless, we would like to argue that pruning itself might not have the flexibility to change the scale since it only involves the operation of removing weights. From this perspective, we believe that our algorithm is to align labels with NTK eigenspectrum rather than changing the scale. Indeed, we provide the training loss curve for both SNIP and GraSP in Figure 2, and we can observe that models pruned by GraSP converge much faster than SNIP and also achieve lower training error. To further verify if the difference is caused by using too small a learning rate for SNIP, we conducted experiments with the same setting as in Figure 2, but increased the learning rates for SNIP. We tried learning rates of 0.3, 1.0 and 2.0. The final test accuracies are:
> +------------------------------------------------------------------------+
> |   LR   |           0.3         |           1.0         |           2.0          |
> +------------------------------------------------------------------------+
> |   Acc |  55.5(+/- 1.2)  |  48.7(+/- 1.6)  |  10.95(+/- 6.9) |
> +------------------------------------------------------------------------+
> All experiments are averaged over three runs. These results show that further performance gain cannot be obtained by simply using larger learning rates.  The corresponding training loss curve can be viewed in https://drive.google.com/file/d/1KUcsGhgj9p1X7rPa7v_0_D5JEWTtVjOR/view . Overall, increasing the learning rate for SNIP does NOT result in better final accuracy or accelerated optimization.
>
> (Minor: Precisely, NTK has the same eigenspectrum as the empirical Fisher matrix rather than the Hessian matrix, though in some cases they are equivalent.)
>
> (2) We have observed that, for large pruning ratios, underfitting is indeed the major problem for pruning algorithms such as SNIP (indicated by the fact that final training error is far away from 0, see Figure 2), because the capacity of the pruned network will be largely affected by the resulting structure, and SNIP will in general result in a bottleneck (prune too many weights) in intermediate layers, whereas it is less severe for GraSP. Besides, we would argue that the bad performance of SNIP for high pruning ratios is not due to a lower effective learning rate based on our results reported in (1) (see above). We didn't observe clear performance improvement by tuning the learning rates for SNIP. Therefore, it’s much more likely that the bad performance of SNIP is due to the pruning strategy induced by the SNIP objective.
>
> (3) No, the batch size is only for the computation of the Hessian vector product in the GraSP. The training procedure is the same as stated in sec 5.1.
>
> We hope our response resolves your concerns well, and if you have any further questions or concerns, please let us know!

---

> > ### Comment · AnonReviewer2 · 2019-11-15
> > **Thank you for your rebuttal**
> >
> > Thank you for your rebuttal.
> >
> > Ad 1) I agree that GraSP shouldn't increase the scale. But my argument was actually mainly just to highlight a weird implication of the theory. The theoretical argument is equivalent to saying that large gradient norm is desirable, but this is clearly not true generally speaking; gradient explosion is not desirable in the general case.
> >
> > I think the argument should be constructed around a standard metric in optimization such as conditioning of the Hessian. The norm of the NTK kernel is clearly not a standard way to argue for an optimization benefit. Maybe GraSP doesn't change the scale, but reduces gradient confusion, or other related metric? Or maybe it improves conditioning of the NTK kernel? Either argument would be more convincing from the optimization perspective.
> >
> > Ad 2) Thank you for checking different learning rates.
> >
> > If the primary reason that GraSP outperforms SNP at high prunning ratios is that SNP underfits, I am not sure this is a novel enough contribution. Perhaps there are some simple heuristics that would reduce change that SNP picks some critical connections?
> >
> > I am not an expert in the field. In the case Reviewer #4 thinks that this is a strong enough contribution, I would be OK accepting the submission.

---

> > > ### Author Response · Authors · 2019-11-15
> > > **Further Response**
> > >
> > > Thank you for your reply and continuing effort to provide constructive feedback until the end of the response/discussion phase.
> > >
> > > 1) Because GraSP shouldn’t increase the scale, so increasing the gradient norm is more likely to be achieved by aligning larger eigenvalues of the NTK with the target. We strongly agree with the reviewer that large gradient norm might not be desirable, but it can typically be handled with small learning rate. If the reviewer is comfortable with our statement about NTK, we will modify it in our camera-ready version if our paper gets accepted.
> > >
> > > 2) The goal of pruning is to reduce the model size while with minimal loss in test accuracy, it does not matter the resulting model overfits or underfits the training data. Therefore, the common strategy for comparing pruning algorithms is to compare the trade-off between model size and test accuracy achieved by each algorithm. Also, underfitting is common for high pruning ratios, but different pruning algorithms always result in drastically different empirical performance, and usually better algorithm will achieve better performance. For example, algorithms such as  DSR, SET and DeepR, they all underfit the training data for pruning ratio 98% on CIFAR10 with ResNet32, but DSR can achieve significantly better results than the other two. We totally agree we can design heuristics to improve all of these pruning algorithms.
> > >
> > > -------------------------------------------------------------------------------------------------------------------------------------
> > >
> > > Finally, we argue that single-shot pruning is promising and offers a new way to speed up network training and inference. This area is new but we believe it will be a very impactful research direction.
> > > More importantly, unlike other traditional pruning algorithms, it has a deep connection with neural network training dynamics and our work may be of independent interest for deep learning theory community. Particularly, large gradient norm indicates big stiffness/gradient confusion (assuming we don't change the scale, see the following references), which seems to correlate with good generalization performance across different tasks.
> > >
> > > https://arxiv.org/pdf/1907.07287.pdf
> > > https://arxiv.org/abs/1901.09491
> > > https://openreview.net/pdf?id=ryeFY0EFwS
> > >
> > > Another potential but promising application of single-shot pruning is to select a big winning ticket (with similar size of standard neural networks) from a gigantic network which cannot fit in our hardware for training. As shown by recent papers in deep learning theory, over-parameterization leads to better generalization, therefore the winning ticket from a gigantic network may perform better than standard neural networks of same size.

---

### Official Review · AnonReviewer1 · 2019-10-23
**Official Blind Review #1**

**Rating:** 6

**Review:**

This paper introduces a method to prune networks at initialization in a way that (mostly) preserves the gradient flow through the resulting pruned network. This is a direct improvement over previous methods (e.g. SNIP) which have no guarantees that pruned connections will break the gradient flow and thereby harm learning.

I quite like this paper, the motivation and results are convincing and it is well presented. The writing is excellent for most of the paper. From section 5 onwards the writing does need quite a bit of editing, as its quality is significantly reduced from what came before.

Some detailed comments:
- Figure 1 is very nice and really clarifies the idea!
- In paragraph below Equation (8): what does "can be computed by backward twice" mean?
- Please specify where the equalities in equation (9) are coming from.
- Table 3 & 4: Why are the pruning ratios different for each model?
- Table 3: Why are values missing for the baseline for 80% and 90%?
- Section 5.2: "We observed that, the main bottleneck or pruned... when deriving the pruning criteria": it's not clear where this conclusion is coming from.
- Table 5 has no batch size results, even though you're referencing them in the text.

And some minor comments to help with the writing:
- Intro: "As shown in Dey et al. (2019) that with pre-specified sparsity, they can achieve" would read better as "As shown by Dey et al. (2019), with pre-specified sparsity one can achieve"
- Equation (3): Clarify that this is a function of $t$
- Sentence below Equation (6): "of the pruned network, and thus our goal" remove the "and thus"
- Table 1: Specify that you're reporting accuracy.
- Section 4.1: "e.g. wide ResNet (Zagaruyko & Komodakis, 2016), and thus we can regard" remove the "and thus"
- Sentence below equation (9): "encouraging the eigenspace of \Theta align" add a "to" before "align"
- Sentence before section 5: "it will encourage the eigenspace of the NTK distributing large eigenvalues in the direction of Y, which will in turn accelerates the decrease of the loss (Arora et al., 2019) and benefits to the optimization in A" would read better as "it will encourage the eigenspace of the NTK to distribute large eigenvalues in the direction of Y, which in turn accelerates the decrease of the loss (Arora et al., 2019) and benefits the optimization in A"
- Throughout section 5, write it in present tense rather than past tense. e.g. "In this section, we conduct various experiments" instead of "In this section, we conducted various experiments"
- Sentence below table 2: you have "the the"
- Second paragraph of section 5.1: "We can observe GraSP outperform random pruning clearly" would read better as "We can observe GraSP clearly outperforms random pruning"
- Second paragraph of section 5.1: "In the next, we further compared" remove "In the next"
- Second paragraph of section 5.1: "Besides, we further experimented with the late resetting" remove "Besides"
- Paragraph above section 5.2: "GraSP surpassing SNIP" use "surpasses" instead
- Paragraph above section 5.2: "investigate the reasons behind in Section 5.2 for promoting better understanding" would read better as "investigate the reasons behind this in Section 5.2 for obtaining a better understanding"
- Section 5.2: "We observed that, the main bottleneck" -> "We observe that the main bottleneck"
- Section 5.2: "Besides, we also plotted the the gradient norm of the pruned", remove "Besides" and the extra "the"
- Section 5.2: "the average of the gradients of the entire dataset" use "over the entire dataset"
- Section 5.2: "hopefully more training progress can make as evidenced" would read better as "hopefully more training progress can be made as evidenced"
- Section 5.3 title would be better using "Visualizing" instead of "Visualize"
- Section 5.3: Join the first two sentences with a comma into a single sentence.
- Section 5.3: "In contrast, SNIP are more likely" -> In contrast, SNIP is more likely"
- Section 5.4: "for ablation study" would read better as "via ablations"
- Section 5.4: "we tested GraSP with three different initialization methods;" use a ":" instead of ";"
- Section 6: "Besides, readers may notice that", remove the "Besides"
- Section 6: "traditional pruning algorithms while still enjoy the cheaper training cost. As an evidence," would read better as "traditional pruning algorithms while still enjoying cheaper training costs. As evidence,"
- Your citation for Evci et al. (2019) is missing the publication venue/arxiv ID.

**Experience Assessment:**

I have read many papers in this area.

**Review Assessment: Checking Correctness Of Derivations And Theory:**

I assessed the sensibility of the derivations and theory.

**Review Assessment: Checking Correctness Of Experiments:**

I carefully checked the experiments.

**Review Assessment: Thoroughness In Paper Reading:**

I read the paper thoroughly.

---

> ### Author Response · Authors · 2019-11-09
> **Response to reviewer #1**
>
> Thanks for your detailed comments, and in particular for your valuable suggestions on improving the writing. We've updated our paper to incorporate your suggestions.
>
> Responses to questions/comments:
>
> (1) In paragraph below Equation (8): what does "can be computed by backward twice" mean?
>
> “Backward twice” means that we first compute the gradient with respect to the weights as $\mathbf{g} = \partial \mathcal{L}/\partial \mathbf{\theta}$ (the first backward), and then we compute the Hessian vector product by simply computing $\mathbf{Hv} = \partial (\mathbf{g}^\top \mathbf{v})/\partial \mathbf{\theta}$ (the second backward, and we only differentiate through $\mathbf{g}$). By doing so, we do not need to compute the Hessian explicitly.
>
> (2) Please specify where the equalities in equation (9) are coming from.
>
> First of all, $\nabla \mathcal{ L}(\mathbf{\theta}) =  \nabla_\mathbf{\theta} \mathcal{Z}^\top \nabla_\mathcal{Z} \mathcal{L}$, where $\mathcal{Z}$ is defined in sec 2.2, page 3. Then we can rewrite $\nabla \mathcal{L}(\mathbf{\theta})^\top \nabla \mathcal{L}(\mathbf{\theta})$ as the second term in equation (9). As we reviewed in sec 2.2 (the paragraph below equation (3)), we can decompose the NTK $\Theta$ as $\sum_{i=1}^n\lambda_i\mathbf{u}_i\mathbf{u_i}^\top$, and plug it in the equation (9) can show the equality.
>
> (3) Table 3 & 4: Why are the pruning ratios different for each model?
>
> The choice of pruning ratios depends on the specific dataset and base network. For ImageNet, we cannot prune as extreme as on Tiny-ImageNet, otherwise the performance of the pruned network will degrade too much and making the comparisons not meaningful. For ResNet32, it is already much more compact than VGG19, so we need to use smaller pruning ratios for ensuring the comparisons are meaningful.
>
> (4) Table 3: Why are values missing for the baseline for 80% and 90%?
>
> It’s not missing. The baseline is the unpruned network (pruning ratio = 0%), rather the sub-network corresponding to pruning ratios of 60%, 80% or 90%.
>
> (5) Section 5.2: "We observed that, the main bottleneck or pruned... when deriving the pruning criteria": it's not clear where this conclusion is coming from.
>
> The observation comes from Figure 2. We can see that the training error of SNIP-pruned network is far away from 0, which means it cannot fit the training data well, i.e. underfitting.
>
>
> We hope our response can address your concerns well. If you have any further questions or concerns, please let us know!

---

### Official Review · AnonReviewer4 · 2019-10-31
**Official Blind Review #4**

**Rating:** 6

**Review:**

This paper proposes a novel one-shot-pruning algorithm which improves the training of sparse networks by maximizing the norm of the gradient at initialization. The utility of training sparse neural networks and shortcomings of dense-to-sparse algorithms like Pruning, LotteryTicket are nicely motivated at introduction. The pruning criterion is motivated by the first order approximation of the change in the gradient norm when a single connection is removed, though the results show that removing many connections together with GraSP increases the total gradient norm therefore allowing the loss to decrease faster. Experiments suggest employing such pruning algorithm improves final performance over two baselines: random and SNIP.

Though I find the proposed method intriguing and well motivated, experiments section of the paper misses some important sparse training baselines and needs some improvement. I am willing to increase my score given my concerns/questions below are addressed.

(1) The paper doesn't mention some important prior work on the topic. Since the paper focuses on end-to-end sparse training, the following sparse training methods needs to be considered and compared with:
- Scalable training of artificial neural networks with adaptive sparse connectivity inspired by network science [Mocanu, 2018]
- Parameter Efficient Training of Deep Convolutional Neural Networks by Dynamic Sparse Reparameterization [Mostafa, 2019]
- Deep Rewiring: Training very sparse deep networks [Bellec, 2017]
- There is also few recent work submitted to ICLR2020: https://openreview.net/forum?id=SJlbGJrtDB, https://openreview.net/forum?id=ryg7vA4tPB, https://openreview.net/forum?id=ByeSYa4KPS

(2) Pruning baselines can be improved. I am not convinced that they represent the best achievable sparse training results. I would recommend method proposed by `Prune or Not to Prune` as a strong baseline. You can also check `State of Sparsity` paper to obtain some competitive pruning results.

(3) It's great that the authors are aware of the importance of having experiments on larger datasets. Though, I found the results reported on Imagenet to be limited. Is there a reason why Imagenet-2012 results are missing pruning baselines? I think having other reported pruning results here along with performance of other sparse training methods (SET, DSR) would be useful. Most of these numbers should be readily available in the papers mentioned above, but I guess it is always better to run them using the same settings.

(4) To demonstrate the usefulness of the pruning criteria proposed, it would be nice to do some simple ablations. Some suggestions: (1) Remove weights that would *decrease* the gradient norm most (2) Do random pruning while preserving exact per layer sparsity fractions. (3) sweep over batch size used to calculate the importance scores and evaluate final accuracies or the initial gradient norm. The second experiment would help identifying whether the gains are due to better allocation of sparsities across layers or due to increased gradient norm. Looking at Figure-4 and seeing the per layer sparsities are different, It is not clear to me which one is the underlying reason for improved performance.

(5) (Page 8 / Table 5) Do you aggregate all accuracies in Table 5 using different batch sizes and initialization methods? If, so I am not sure what the intended message here is, since it is difficult to infer how these hyper-parameters affect the result. Do you sweep different batch sizes for estimating importance of units, too? It would be nice to see whether the two batch sizes interact with each other and/or how increased batch size affects the quality of pruned networks.

Some minor comments:
(a) (Page 1) I found the motivation very intriguing. Though the statement `Recently, F&C (2019) shed light on this...` seems a bit off, given that LT can't find solutions as well as the pruning solution in most practical (larger datasets and architectures) settings. Therefore I would be better to pose this as an `open problem`.

(b) (end of page-1) `However, connection sensitivity is sub-optimal as a criterion because the gradient of each weight might change dramatically after pruning due to complicated interactions between weights`. I think this is still the case for GraSP. Since the criterion it uses assumes independence (i.e. what if we remove a single weight?). It would be nice to see some ablations on this. Does `K=number of weights removed` affect the norm of the sparsified networks?

(c) (Figure 1) I find the comparative illustration between SNIP and GraSP very useful. Though, the architecture presented seems a bit artificial (i.e. I am not aware of any architecture with single hidden layer and a single output unit). I think the same motivation can be made by removing the top unit (therefore having 6-4-1 units) and removing all incoming connections for the output unit until a single connection remains. Then SNIP would remove that single connection whereas GraSP would remove one of the connections in the previous layer.

(d) (Section 2.1) `In contrast, Hessian based algorithms...` Though it is a structured pruning algorithm It might be nice to include the following work, https://arxiv.org/abs/1611.06440.

(e) (Section 2.1) Previous work needs following citations:  [Bellec, 2017], [Mocanu, 2018] and [Mostafa, 2019]

(f) (Section 2.2) Why the initial dynamics affect the final performance? One explanation given in the paper is through recent work on NTK and this is great. Though training settings used at `Lee et al (2019a)` and in the paper are a bit different. Usage of MSE, small datasets, etc… So it might be nice to point out differences.

(g) (Section 3) At $D = {(x_i, y_i)}_{i=1}^n$, `n`->`N`

(h) (Page 4) `Preserving the loss value motivated several…` -> `motivated by several…`
I think it is better to use existing terminology whenever available.I think using `One-shot pruning` instead of `Foresight pruning` would be a better choice and would prevent confusion.

(j) (Page 5) `However, it has been observed that different weights are highly coupled …` This has been observed much earlier, too: like in Hassibi, 1993.

(k) (Page 7) Last sentence `and thus hopefully..`: needs to be fixed.

(l) (Page 8) The whole page needs some proof-reading. Some of them: (a) `SNIP and GraSP. We present...` probably connect with comma (b) `aims for preserving` -> `aims to preserve` (c) `In contrast, SNIP are more` `are`->`is` (d) `for ablation study` -> `as an ablation study`...

(m) Is there a specific reason why VGG networks are preferred for experiments? I don't think they are relevant to any practical application anymore and they are massively over-parameterized for tasks in hand. Specifically for Cifar-10. I think focusing on more recent networks and larger datasets would increase the impact of the work.

**Experience Assessment:**

I have published one or two papers in this area.

**Review Assessment: Checking Correctness Of Derivations And Theory:**

I assessed the sensibility of the derivations and theory.

**Review Assessment: Checking Correctness Of Experiments:**

I carefully checked the experiments.

**Review Assessment: Thoroughness In Paper Reading:**

I read the paper thoroughly.

---

> ### Author Response · Authors · 2019-11-14
> **Response to reviewer #4 [4/4]**
>
> (3) ImageNet baselines;
>
> Thank you for your kind words, we strongly agree with you that large scale experiments are important and necessary. Our purpose of ImageNet experiments is only for showing that GraSP can beat SNIP consistently even on more challenging and larger datasets. As we mentioned in the beginning of our response, we think the only baseline of single-shot pruning is SNIP. Therefore, we did not include other baselines in this experiment. We agree that including more baselines will make our empirical results stronger, but it won’t change our conclusion that GraSP is better than SNIP.  To have a sense, we provide a rough comparison between the results of SET, DSR, Deep-R and GraSP on ImageNet with ResNet50 referred from their original papers:
>
> +———---+——————+——————+——————--+———-——--+
> |   Model   |       SET         |        DSR        |      Deep-R      |      GraSP      |
> +———---+——————+——————+——————--+—————---+
> |   80%     |        72.6        |        73.3        |         71.7         |       72.06       |
> +———---+——————+——————+——————--+—————---+
> |   90%     |        70.4        |        71.6        |         70.2         |       68.14       |
> +———---+——————+——————+——————--+—————---+
> We can observe that GraSP is still quite competitive in this setting, and it outperforms DeepR at the pruning ratio of 80%, though GraSP is a single-shot pruning algorithm. It is very encouraging that single-shot pruning algorithm can perform as competitively as other ‘Pruning during Training’ methods.
>
>
> (4) Usefulness of the pruning criteria.
>
> We really thank reviewer for proposing some interesting ablation studies. (1) For reducing gradient norm, we found that it will result in disconnected networks for high pruning ratios, and thus correspondingly performs much worse. (2) For ‘random pruning’, we adopt the sparsity allocation identified by GraSP and then shuffle the sparse mask. We found that for low pruning ratios, shuffling the mask does not degrade the performance much, while for high pruning ratios, i.e., 98%, 99%, it will degrade the performance a lot. We conjecture that for low pruning ratios, the pruned network is still moderately over-parameterized, and thus the shuffling operation will not affect the performance much. Apart from these ablation study, we believe that the best way for showing the usefulness of a pruning criteria is the empirical results in terms of pruning-ratio vs. Test accuracy.
>
> (5) (Page 8 / Table 5) Do you aggregate all accuracies in Table 5 using different batch sizes and initialization methods?
>
> Yes, they are averaged over multiple runs. The purpose of them is for sensitivity analysis, so as to show that our pruning criteria is not sensitive to different batch sizes and initialization schemes.
>
> - Response to minor comments.
>
> We've updated our paper to incorporate your suggestions on writing and citations. In terms of the reason we reported the results of VGG networks, our main purpose is to simulate the case of feedforward networks without skip-connections (we also reported results on ResNet (i.e. with skip-connections) in our paper). We agree that experimenting with more recent networks is good, but we should avoid doing duplicated experiments.
>
> We really appreciate your valuable comments, and careful assessment of our work. We hope our response can address your concerns well, and if you have any further concerns/questions/suggestions, please let us know!

---

> ### Author Response · Authors · 2019-11-14
> **Response to reviewer #4 [3/4]**
>
> # ResNet32 on TinyImageNet
> +-----------+----------------+----------------+----------------+
> |   Ratio  |        85%      |        90%      |       95%      |
> +-----------+----------------+----------------+----------------+
> |    DSR   |      57.08      |      57.19     |       56.08     |
> +-----------+----------------+----------------+----------------+
> |     SET   |      57.02      |      56.92     |       56.18     |
> +-----------+----------------+----------------+----------------+
> |  DeepR |      53.29      |      52.62     |       52.00     |
> +-----------+----------------+----------------+----------------+
> |  Grasp  |  57.25(0.1)  | 55.53(0.1)  |  51.34(0.3)  |
> +-----------+----------------+----------------+----------------+
>
> # VGG19 on TinyImageNet
> +-----------+----------------+----------------+----------------+
> |   Ratio  |        90%      |        95%      |       98%      |
> +-----------+----------------+----------------+----------------+
> |    DSR   |      62.43      |      59.81     |       58.36     |
> +-----------+----------------+----------------+----------------+
> |     SET   |      62.49      |      59.42     |       56.22     |
> +-----------+----------------+----------------+----------------+
> |  Grasp  |  60.76(0.2)  | 59.50(0.3)  |  57.28(0.3) |
> +-----------+----------------+----------------+----------------+
> (Note: DeepR is missing in the above table because it is extremely slow. We will complete once the experiments are finished.)
>
> Further discussions for the above results:
>
> We can observe that DSR is the best performing one, and GraSP is quite competitive. In particular, GraSP performs much better than DeepR in most settings, and can outperform SET in more than half of the settings. Furthermore, we would also argue that GraSP has several advantages over these three ‘Pruning during Training’ methods:
>
>  - Simplicity and easy-to-use: GraSP is much simpler than DSR, SET and DeepR, as it only needs to conduct pruning prior to training in a single-shot, and it does not change the sparsity dynamically during training. Moreover, there is almost no hyperparameters to tune for GraSP in comparisons with DSR, SET and DeepR. (i.e. DeepR is extremely slow and not scalable; SET requires manually specified pruning ratio for each layer; DSR requires some specific layers to be dense.)
>
>  - Efficiency: GraSP can enjoy training acceleration (5x) by optimization in the hardware level (i.e. mapping the network topology structure to circuits or FPGA pre-programmed wiring). As we mentioned in [A], Dey et al. (2019) show that  with **pre-specified** sparsity, the training can be accelerated by 5x. However, for dynamic sparse training (DSR, SET and DeepR), they need to change the sparse mask during training and thus cannot be optimized in the hardware level, and there is no GPU-accelerated libraries that utilize sparse tensor exist (Dettmers & Zettlemoyer 2019). Also it is almost impractical to optimize the training efficiency of dynamic sparse method in the hardware level, because their topology will change during training and recompile the FPGA program or changing the circuits will make the training even slower.
>
> In general, we will not be surprised if DSR, SET, DeepR and other ‘Pruning during Training’ methods outperform ‘Pruning before Training’ methods, as they can change the sparsity during training dynamically and thus enjoy more flexibility than GraSP and SNIP. But they also have the disadvantages as we stated in A.a, A.b and the previous paragraph.
>
> The baselines contained in ‘State of the sparsity ’ are all in the category of ‘Pruning after Training’, and they all require a pretrained network, which does not save the training cost and cannot be directly compared with GraSP. In our paper, these methods are only used to serve as an upper bound for ‘Pruning before Training’ methods, and we already included two of them, OBD and MLPrune. Moreover, in the ‘State of the sparsity’ paper (Gale et al., 2019), they did not include Hessian-based pruning algorithms, such as OBD and MLPrune, into their comparisons with magnitude pruning. We also find that in another ICLR submission ( https://openreview.net/pdf?id=ryl3ygHYDB ), they have demonstrated that OBD can significantly outperform magnitude pruning (See Table 19 and Table 20), which is the best performing one in the ‘State of the sparsity’ paper. Besides, we would also argue that SNIP is the most related, state-of-the-art baseline for pruning network prior to training.
>
> ----------------------------------------------------------------------------------------------------------------------------------------------------------------------------
> Gale, Trevor, Erich Elsen, and Sara Hooker. "The state of sparsity in deep neural networks." arXiv preprint arXiv:1902.09574 (2019).

---

> ### Author Response · Authors · 2019-11-14
> **Response to reviewer #4 [2/4]**
>
> (1-2) Include more baselines. (DSR, SET, DeepR ); Pruning baselines can be improved. I am not convinced that they represent the best achievable sparse training results.
>
> We really thank the reviewer for pointing out these sparse training papers and related concurrent submissions. We have updated the paper to include them in the section of related works. We also agree with the reviewer that including those sparse training methods, DSR, SET and DeepR, will greatly improve the experiments section. Therefore, we adopt the public implementation from https://github.com/IntelAI/dynamic-reparameterization for the experiments with DSR, SET and DeepR (Bellec et al., 2018). Specifically, we test them on three datasets (CIFAR-10, CIFAR-100 and TinyImageNet) with two networks (VGG19 and ResNet32). The results are presented in the following:
>
> # ResNet32 on CIFAR10
> +-----------+----------------+----------------+----------------+
> |   Ratio  |        90%      |        95%      |       98%      |
> +-----------+----------------+----------------+----------------+
> |    DSR   |      92.97      |      91.61     |       88.46     |
> +-----------+----------------+----------------+----------------+
> |     SET   |      92.30      |      90.76     |       88.29     |
> +-----------+----------------+----------------+----------------+
> |  DeepR |      91.62      |      89.84     |       86.45     |
> +-----------+----------------+----------------+----------------+
> |  Grasp  |  92.38(0.2)  | 91.39(0.3)  |  88.81(0.1)  |
> +-----------+----------------+----------------+----------------+
>
> # ResNet32 on CIFAR100
> +-----------+----------------+----------------+----------------+
> |   Ratio  |        90%      |        95%      |       98%      |
> +-----------+----------------+----------------+----------------+
> |    DSR   |      69.63      |      68.20     |       61.24     |
> +-----------+----------------+----------------+----------------+
> |     SET   |      69.66      |      67.41     |       62.25     |
> +-----------+----------------+----------------+----------------+
> |  DeepR |      66.78     |      63.90     |       58.47     |
> +-----------+----------------+----------------+----------------+
> |  Grasp  |  69.24(0.2)  | 66.50(0.1)  |  58.43(0.4) |
> +-----------+----------------+----------------+----------------+
>
> # VGG19 on CIFAR10
> +-----------+----------------+----------------+----------------+
> |   Ratio  |        90%      |        95%      |       98%      |
> +-----------+----------------+----------------+----------------+
> |    DSR   |      93.75      |      93.86     |       93.13     |
> +-----------+----------------+----------------+----------------+
> |     SET   |      92.46      |      91.73     |       89.18     |
> +-----------+----------------+----------------+----------------+
> |  DeepR |      90.81      |      89.59     |       86.77     |
> +-----------+----------------+----------------+----------------+
> |  Grasp  |  93.30(0.1)  | 93.04(0.2)  |  92.19(0.1)  |
> +-----------+----------------+----------------+----------------+
>
> # VGG19 on CIFAR100
> +-----------+----------------+----------------+----------------+
> |   Ratio  |        90%      |        95%      |       98%      |
> +-----------+----------------+----------------+----------------+
> |    DSR   |      72.31      |      71.98     |       70.70     |
> +-----------+----------------+----------------+----------------+
> |     SET   |      72.36      |      69.81     |       65.94     |
> +-----------+----------------+----------------+----------------+
> |  DeepR |      66.83     |      63.46     |       59.58     |
> +-----------+----------------+----------------+----------------+
> |  Grasp  |  71.95(0.2)  | 71.23(0.1)  |  68.90(0.4) |
> +-----------+----------------+----------------+----------------+
>
> ----------------------------------------------------------------------------------------------------------------------------------------------------------------------------
> - Guillaume Bellec , David Kappel, Wolfgang Maass, and Robert Legenstein"Deep Rewiring: Training very sparse deep networks." ICLR 2018.

---

> ### Author Response · Authors · 2019-11-14
> **Response to reviewer #4 [1/4]**
>
> Thanks for your detailed reviews. We really appreciate your time for reviewing our paper carefully.
>
> Before moving on to answer your questions and comments, we’d like to first clarify the focus of this work. Our goal is to propose an improved pruning algorithm which can conduct pruning before training. In this sense, we don’t think those sparse training baselines relevant enough. To the best of our knowledge, the only existing baseline is SNIP (Lee et al., 2019). Other algorithms such as OBD (LeCun et al., 1990) and MLPrune (Zeng & Urtasun, 2019) in our paper are serving as upper bound for single-shot pruning algorithm and the main reason to include them is to inform readers of the gap between pruning before and after training. To address your concerns, we run experiments with the methods you mentioned. However, we note again that those methods only serve as the upper bound and are not really “important” baselines.
>
> [A] Regarding the sparse training baselines you mentioned, we would like to clarify the difference between ‘Pruning before Training’ and ‘Pruning during Training’.
> - (a) ‘Pruning during Training’ methods, such as SET (Mocanu et al., 2018) and DSR (Mostafa & Wang, 2019), need to redistribute the weights during training by different heuristics. As shown in Dey et al. (2019), with *pre-defined sparsity*, the training can be accelerated by 5X, and the speedup performance can be optimized in the hardware level by programming the sparse structure using FPGA in advance of training. However, SET and DSR need to change the sparse structures during the whole training process, and thus it is unclear for those methods to enjoy the potential acceleration from hardwares.
> - (b) ‘Pruning during Training’ methods change the standard training procedure because they need to redistribute the weights during training, which makes it more complicated than ‘Pruning before Training’.
> - (c) ‘Pruning during Training’ methods enjoy more flexibility than ‘Pruning before training’ as they have the freedom to change the sparsity during training. However, this results in the problem of (a) and (b).
> For other detailed differences, please refer to the ICLR2020 submission at: https://openreview.net/pdf?id=SJem8lSFwB
>
> ----------------------------------------------------------------------------------------------------------------------------------------------------------------------------
> - Lee, Namhoon, Thalaiyasingam Ajanthan, and Philip HS Torr. "Snip: Single-shot network pruning based on connection sensitivity." ICLR 2019.
> - LeCun, Yann, John S. Denker, and Sara A. Solla. "Optimal brain damage." Advances in neural information processing systems. 1990.
> - Zeng, W. and Urtasun, R. MLPrune: Multi-layer pruning for automated neural network compression, 2019. URL https://openreview.net/forum? id=r1g5b2RcKm.
> - Mocanu, Decebal Constantin, et al. "Scalable training of artificial neural networks with adaptive sparse connectivity inspired by network science." Nature communications 9.1 (2018): 2383.
> - Mostafa, Hesham, and Xin Wang. "Parameter efficient training of deep convolutional neural networks by dynamic sparse reparameterization." ICML 2019.
> -Sourya Dey, Kuan-Wen Huang, Peter A Beerel, and Keith M Chugg. "Pre-defined sparse neural networks with hardware acceleration." IEEE Journal on Emerging and Selected Topics in Circuits and Systems, 2019.

---

> ### Comment · AnonReviewer4 · 2019-11-15
> **Few Clarifications Needed**
>
> Naming Dynamic Sparse Training methods as 'Pruning during Training' can be misleading. Since most of the pruning algorithms start with a dense network and prune the networks during training. I would suggest naming those methods as 'Dynamic Sparse Training'(DST) methods. These methods start with a predefined sparsity, but unlike `One Shot pruning` algorithms, they do change the connectivity of layers during training. This has been shown to improve performance over keeping the connectivity static.
>
> "Our goal is to propose an improved pruning algorithm which can conduct pruning before training. In this sense, we don’t think those sparse training baselines relevant enough. "
>
> I disagree with this. Methods come with the problem they address. DST algorithms are very relevant to your algorithm, since they attack the same problem: *Training sparse neural networks* with the same goal: *Reducing training cost*. They would benefit same improvements in terms of FLOPs as GrasP since they do start from a *predefined sparsity* (term taken from Dey et.al. 2019).
>
> I think the comparison you provided for DST and One-shot-pruning algorithms is a great start. Dey et.al.'s paper is very interesting. They talk about hardware friendly, clash-free, sparse connectivity patterns and show that such patterns perform just as good as any random pattern (static training). There is no guarantee that GrasP would find such clash-free connectivity. Please correct me if I am wrong (A). I am not an expert on hardware, but it is also not obvious to me that FPGA's would be the choice of hardware for training in the future(it is currently not). What about CPU, GPU acceleration of sparse neural network training? (B)
>
> Thanks for running extra experiments. I appreciate your work. Results show that GrasP performs worse than some of the DST methods (which is fine). As discussed by the authors there might be some settings where static sparsity is preferred (still not sure about this) and not all papers should get SOTA for the problem they attack. However, they should make a fair comparison with relevant methods.
>
> I don't see any of comparisons and numbers provided below in the revised paper. Are the authors planning adding those results in the next version?  Similarly I believe experiments done for '(4) Usefulness of the pruning criteria' would be a great addition to your work. (C)
>
> A quick response to questions (A), (B), (C) would be helpful.

---

> > ### Author Response · Authors · 2019-11-15
> > **Quick response**
> >
> > Thank you for your reply. We sincerely appreciate your valuable comments.
> >
> > Thank you for correcting us in terms of DST algorithms. We will take your suggestions in our new revision.
> >
> > (A) Currently, we’re not aware of such guarantee. But if the sparsity pattern does not change during training, this might ease the difficulty of designing dedicated hardware for achieve accelerations.
> >
> > (B) It seems that there is no GPU acceleration library for sparse tensors  (Dettmers & Zettlemoyer, 2019). However, we cannot exclude the possibility of it in the future. CPU is in general not capable of dealing with massive computations, and even for sparse networks, it is still requires a large amount of computations.
> >
> > (C) We’re now working on a new revision to include those results and will try our best to update to openreview before the rebuttal deadline.
> >
> > Finally, we’d like to note that our single-shot pruning algorithm may be of independent interest for deep learning theory community. Lottery ticket hypothesis (they showed that there exist winning tickets at the initialization) paper opened a new line of research in understanding neural network dynamics. However, in the original lottery ticket hypothesis paper, they had to use *pruning after training* method to identify the ticket. Our paper is trying to show that we are able to identify them before training and push it to the limit. We believe it may have some inspirations for other researchers in understanding deep learning and neural network training.
> >
> > We also believe our pruning criteria has some deep connections with generalization performance of neural networks. Particularly, large gradient norm indicates big stiffness/gradient confusion (assuming we don't change the scale, see the following references), which seems to correlate with good generalization performance across different tasks.
> >
> > https://arxiv.org/pdf/1907.07287.pdf
> > https://arxiv.org/abs/1901.09491
> > https://openreview.net/pdf?id=ryeFY0EFwS
> >
> >
> > ------------------------------------------------------------------------------------------------------------------------------
> > Tim Dettmers and Luke Zettlemoyer. Sparse networks from scratch: Faster training without losingperformance.arXiv preprint arXiv:1907.04840, 2019.

---

> > ### Author Response · Authors · 2019-11-15
> > **We've updated the paper.**
> >
> > We've updated the paper to include three DST baselines and separate DST methods from *Pruning during training* ones in related works. However, we haven't finished the experiments of DeepR on Tiny-ImageNet. We will add it to our camera-ready version if our paper gets accepted.
> >
> > We thank again for your detailed and constructive comments.

---

### Author Response · Authors · 2019-11-14
**To all reviewers**

We thank all reviewers again for your detailed comments and constructive suggestions.

We've updated our paper and responded to you. We really sorry for the late response to reviewer #4, it took many days for us to run experiments you requested. We hope that our responses address your concerns. If so, it would be great if you can update your review and rating. But if not, we're open to answer more questions and further improve our paper

---

### Decision · Program_Chairs · 2019-12-19

**Decision:**

Accept (Poster)

**Comment:**

This paper proposes a method to improve the training of sparse network by ensuring the gradient is preserved at initialization. The reviewers found that the approach was well motivated and well explained. The experimental evaluation considers challenging benchmarks such as Imagenet and includes strong baselines.